# Poly(Aspartic Acid) Promotes Odontoblast-like Cell Differentiation in Rat Molars with Exposed Pulp

**DOI:** 10.3390/jfb14110537

**Published:** 2023-11-01

**Authors:** Fernanda Furuse Ventura dos Santos, Stefan Habelitz, Fábio Dupart Nascimento, Victor Elias Arana-Chavez, Roberto Ruggiero Braga

**Affiliations:** 1Department of Biomaterials and Oral Biology, School of Dentistry, University of São Paulo, São Paulo 05508-000, SP, Brazil; vearana@usp.br (V.E.A.-C.); rrbraga@usp.br (R.R.B.); 2Department of Preventive and Restorative Dental Sciences, School of Dentistry, University of California, San Francisco, CA 94143, USA; stefan.habelitz@ucsf.edu; 3Molecular Biology Division, Department of Biochemistry, Federal University of São Paulo, São Paulo 04044-020, SP, Brazil; fabio.dupart@unifesp.br

**Keywords:** dental pulp capping, tooth remineralization, calcium silicate, dentin regeneration

## Abstract

In recent years, alternative pulpal therapies targeting dentinogenesis signaling pathways using different peptides have been investigated. The aim of this study was to verify the effectiveness of poly(aspartic acid), pAsp, in dentin regeneration using an animal model. Methods: Mechanical pulp exposure was performed in the upper molars of 56 Wistar rats, randomly divided as follows (n = 14): control (no treatment); MTA group—pulp capping with mineral trioxide aggregate (MTA Angelus); pAsp group—application of 20 μL of pAsp solution (25 mg·mL^−1^); MTA+pAsp group—application of MTA mixed with pAsp (5:1 by mass). Animals were euthanized after 7 or 21 days. Histological sections were submitted to hematoxylin-eosin and Brown and Brenn staining and immunohistochemical analysis for osteopontin (OPN) and dentin matrix protein 1 (DMP 1). Results: At 7 days, an acute inflammatory infiltrate and the presence of disorganized mineralized tissue were observed in all groups. At 21 days, the quality and thickness of the reparative dentin in treated groups were superior to the control, and bacterial contamination was observed in two MTA-pAsp specimens. While all treated groups showed intense immunostaining for OPN at 21 days, only the pAsp group expressed DMP 1, indicating the presence of fully differentiated odontoblast-like cells. Conclusion: Poly(aspartic) acid promoted dentin regeneration in rat molars in the absence of an additional calcium source and may be an alternative to MTA as a pulp-capping agent.

## 1. Introduction

The repair of exposed pulp tissue and the formation of reparative dentin require the application of bioactive materials to the exposure site to stimulate cell proliferation and differentiation [1,2]. Reparative dentin is devoid of dentinal tubules and usually contains cellular inclusions, which resemble osteocytes from bone, thus exhibiting a primary bone appearance. In addition, the reparative dentin secreted by differentiating odontoblasts contains numerous bone proteins in its extracellular matrix, including large amounts of osteopontin (OPN), a non-collagenous protein scarce in dentin but abundant in primary bone. Contrarily, some typical dentin proteins, such as dentin matrix protein 1 (DMP 1), are absent at the early stages of reparative dentin formation. Only when fully differentiated odontoblasts are capable of secreting specific dentin proteins [3].

For decades, calcium hydroxide cement, Ca(OH)_2_, was considered the “gold standard” for pulp capping; however, cements based on calcium silicate, such as mineral trioxide aggregate (MTA), initially introduced for the treatment of root perforations and retrofilling in endodontic surgeries, have been widely used as an alternative to Ca(OH)_2_ [4,5]. Although there is not enough clinical data to assert the superiority of one material over the other [6,7,8], in general, in vitro studies point to a better performance of materials based on calcium silicate. Calcium silicate cements maintain their high alkaline pH over a longer period of time than Ca(OH)_2_. With this, the antimicrobial effect is prolonged, contributing to the control of inflammation [5,9]. Additionally, the mineralized barrier formed with calcium silicate-based cements has fewer vascular inclusions and defects and greater thickness [10,11]. Furthermore, MTA has osteoinductive activity and the ability to bind to the mineralized bone matrix [12].

Clinical studies show a reduction in the success rates of direct pulp capping in the long term [13,14,15], which is explained, in part, by the fact that pulp capping materials do not have specific control over the cell signaling mechanisms that would make it possible to restore the structural and functional characteristics of dentin [16]. As a consequence, thin mineralized barriers (increasing the risk of recontamination of adjacent tissue) or an excess of disorganized mineralized tissue are frequent findings in histological evaluations [10,17].

Recently, strategies based on the modulation of dentinogenesis signaling pathways for dentin regeneration have been investigated to overcome the limitations of traditional therapies and improve long-term prognosis. For example, the upregulation of Wnt/β-catenin through the application of drugs that inhibit the enzyme glycogen synthase kinase 3 (GSK-3) in pulpal exposure areas of mouse molars increased the mobilization of stem cells and their differentiation into odontoblast-like cells, resulting in the formation of reparative dentin with a composition close to native dentin after four weeks [18]. Semaphorin-3A, a protein involved in osteoblastic bone formation through Wnt/β-catenin signaling [19], was shown to induce the formation of reparative dentin with dentinal tubules and a well-aligned layer of odontoblast-like cells in the region of exposed pulp tissue [20].

Different peptides derived from proteins involved in dentinogenesis have been evaluated in dentin regeneration. These peptides were able to promote stem cell recruitment and reparative dentin formation through positive signaling of the NF-kB pathway and inhibition of the MAPK pathway, with results similar to those found with the use of mineral trioxide aggregate (MTA) [21,22,23]. In addition to activating cell signaling pathways, peptide analogs of non-collagenous proteins were shown to stimulate cell differentiation through mechanotransduction in bone. For example, in the guided mineralization of collagen scaffolds, OPN induced the differentiation of mesenchymal stem cells into osteocytes, resulting in the formation of tissue with composition and ultrastructural organization comparable to that of human bone [24].

Poly(aspartic acid), pAsp, is a highly versatile, biodegradable, and biocompatible anionic polypeptide with a high affinity for calcium ions due to the presence of carboxyl groups [25,26]. Indeed, domains with multiple repeats of aspartic acid residues are present in OPN and other non-collagenous proteins found in the matrix of mineralized tissues [27]. A wide range of molecular weights of pAsp has been evaluated for the targeted delivery of bioactive agents, surface modification of biomaterials, and the development of scaffolds for tissue regeneration [25]. Treatment of poly(lactic acid) films with high molecular weight (12 kD) pAsp increased adhesion, proliferation, and differentiation of osteoblasts [28]. Furthermore, Gower et al. observed that the addition of pAsp to a supersaturated solution of calcium and phosphate ions promotes the formation of a liquid nanoprecursor of calcium phosphate mineral linked to the ionizable groups of the molecule [29,30,31], capable of penetrating the intrafibrillar spaces of collagen fibrils [32,33], acting as a directing agent of the mineralization process [34]. In fact, the mechanisms of collagen mineralization induced by pAsp resemble the activity of phosphoproteins in collagen mineralization in vitro, and pAsp may thus mimic the actual biological process of mineral induction in collagen fibrils [35].

This study aimed to evaluate the effectiveness of pAsp associated or not with MTA in dentin regeneration in an experimental animal model. Based on the above and considering its pivotal ability to mimic the natural biomineralization process, this study hypothesized that the presence of pAsp at the pulp exposure site would affect the differentiation behavior of pulpal stem cells and the mineralization of the dentin matrix deposited by odontoblast-like cells, possibly resulting in a better organized mineralized tissue compared to conventional capping therapies.

## 2. Materials and Methods

### 2.1. Experimental Design

Fifty-six Wistar rats (*Rattus novergicus*) with body weights between 250 and 300 g were used. Two factors were evaluated: treatment (in four levels) and observation period (in two levels). Considering the possibility of losing one animal per group and losses due to failure in sample processing, both upper first molars were used in all of the seven animals assigned to each group. Sample size was defined based on previous studies involving dentin regeneration, in which five to eight teeth per group were evaluated [2,36,37,38]. Animals were obtained and kept in the dental school animal house under the following conditions: cycles of 12 h of light and 12 h of dark, room temperature of 22 ± 2 °C, ventilation system/exhaustion allowing 20 air changes per hour, relative humidity around 55 ± 5%, storage in a ventilated rack with plastic cages (four animals per cage), where they had free access to food and water. Animal care and experimental procedures were performed following the university’s Guidelines for Animal Experimentation, with approval from the Ethics Committee for Animal Research (approval statement #07/2020, issued on 8 September 2020).

### 2.2. Anesthesia

Surgical procedures and euthanasia were performed under general anesthesia with ketamine (87 mg/kg, Francotar^®^, Virbac, Carros, France) and xylazine (13 mg/kg, Rompum^®^, Bayer, São Paulo, SP, Brazil).

### 2.3. Mechanical Pulp Exposure

After anesthesia, the tooth was disinfected with cotton pellet soaked in 2% chlorhexidine gluconate, followed by preparation of a cavity on the occlusal surface of the maxillary first molars using a sterile spherical steel bur (1 mm; ln long neck; Dentsply Indústria e Comércio Ltd.a., Petrópolis, RJ, Brazil) coupled to a low-speed handpiece under irrigation with sterile saline solution (0.9% NaCl). The bur was changed every two prepared cavities. Penetration of approximately half the diameter of the active tip of the bur was performed in the mesial fossula of the occlusal surface, and a 1 mm diameter pulp exposure was produced by penetration of a sterile #15 endodontic file. Cavity/exposure size reproducibility was checked using digital photography. Bleeding was controlled using gentle pressure with sterile cotton pellets soaked in saline solution.

### 2.4. Experimental Groups

The 56 animals were randomly divided into two groups according to the postoperative period (7 and 21 days). For each of the postoperative periods, the four treatments were randomized using Microsoft Excel RAND function (Microsoft Corp., Redmond, WA, USA). The same was applied to randomize between the right and left first upper molars of each of the 28 animals selected for that period (n = 14). The following treatments were conducted:(1)Group C (negative control): after the mechanical pulpal exposure procedure, no capping material was applied and a piece of polytetrafluoroethylene (PTFE; Teflon tape—PS65, 3M, Sumaré, SP, Brazil) previously sterilized in autoclave at 121 °C for 40 min was placed at the exposure site to avoid direct contact between the pulp tissue and the coronal restoration made with a flowable resin composite (NT Premium, Vigodent-Coltene, Rio de Janeiro, RJ, Brazil) associated with a two-step adhesive system (Single Bond 2, 3M ESPE, St. Paul, MN, USA);(2)MTA group: pulp capping with mineral trioxide aggregate (MTA-Angelus; Angelus, Londrina, PR, Brazil), mixed with distilled water according to the manufacturer’s instructions. The cement was brought into position using an endodontic probe and accommodated without pressure with the narrow end of a sterile paper cone. The cavity was filled immediately with flowable resin composite, as described for the control group;(3)pAsp group: 20 μL of a solution containing 25 mg/mL of pAsp (27,000 Da, Alamanda Polymers Inc, Huntsville, AL, USA) was applied to the exposure site and left undisturbed for 20 s [34]. The excess solution was removed using sterile cotton balls, and a sterile piece of Teflon tape was placed at the exposure site before sealing the cavity with the resin composite;(4)MTA/pAsp group: MTA particles were mixed with pAsp in a 5:1 ratio (by mass) before the manipulation of the material with distilled water [39]. The material was applied to the pulpal exposure site, and the cavity was sealed as described for the MTA group.

After sealing the cavities, the cusp tips of the antagonist’s teeth were flattened with a low-speed diamond bur to minimize the action of occlusal forces.

### 2.5. Euthanasia and Sample Collection

Euthanasia was carried out 7 or 21 days after the pulp capping, and the hemimaxillae, including the teeth, were carefully dissected. The samples were immersed in a container containing 100 mL of the fixative solution consisting of 4% formaldehyde (freshly prepared from paraformaldehyde) plus 0.1% glutaraldehyde in 0.1 M sodium cacodylate buffer (Sigma, Saint Louis, MO, USA), pH 7.2, placed in a larger glass container filled with ice. The set was immediately placed in a laboratory microwave oven (Pelco 3400, Ted Pella; Redding, CA, USA), where the samples were exposed to microwave irradiation at 100% power for three five-minute cycles under a controlled temperature of 36 °C and then transferred to a new fixative solution and left at 4 °C overnight [40].

### 2.6. Histological Processing

All samples were demineralized in 4.13% ethylenediamine tetraacetic acid (EDTA) in PBS for 45 days and submitted to conventional histological processing for paraffin embedding. Serial longitudinal sections (5 μm thickness) were made in the mesiodistal direction. A minimum number of ten sections were collected per specimen. The histological sections were submitted to hematoxylin-eosin (HE) staining for histopathological evaluation, and Brown and Brenn staining was modified by Taylor to eliminate the possibility of bacterial contamination of the pulp tissue [41]. For immunohistochemical analysis, the indirect immunoperoxidase technique was employed for detection of osteopontin (OPN) and dentin matrix protein 1 (DMP 1).

### 2.7. Histopathological Analysis

Three histological sections from each of the samples were analyzed. Histopathological analysis was performed using an Olympus BX-60 light microscope (Olympus, Tokyo, Japan) coupled to the Cell F image capture system (Olympus, Tokyo, Japan), evaluating the following parameters: presence or absence of bacterial contamination, formation of the mineralized barrier, quality of the mineralized barrier formed, and degree of pulp inflammation.

### 2.8. Immunohistochemical Analysis

For immunohistochemical analysis, the histological sections were divided into two batches and submitted to the indirect immunoperoxidase technique. Histological sections were incubated in the following primary antibodies: mouse monoclonal anti-OPN (SC—21742, Santa Cruz Biotechnology, Santa Cruz, CA, USA) at a concentration of 1:1000 for 3 h and mouse monoclonal anti-DMP 1 (MABD19, Merck KGAA, Darmstadt, Germany) at a concentration of 1:20 for 18 h. For signal amplification, they were incubated in HRP system secondary antibody (Dako EnVision + Dual Link System—HRP, Dako North America, Santa Clara, CA, USA) for 30 min in a dark environment and humid chamber. The reaction was developed using the chromogen diaminobenzidine 0.025% (DAB, 3,3-diaminobenzidine, Dako, Carpinteria, CA, USA) for 10 min, according to the manufacturer’s guidelines. At the end of the immunohistochemical reactions, the sections were counterstained with Mayer’s hematoxylin. The negative control was performed using the protocol described above, without the use of primary antibodies. Immunolabeling was defined as a brownish staining precipitate on cells and/or extracellular matrix. The specificity of the immunolabeling was verified by the absence of staining in the negative controls.

### 2.9. Data Analysis

A qualitative analysis was performed on the sections stained by HE and Brown and Brenn technique modified by Taylor, as well as at the immunolabeled areas at the surgical defect, considering the location, distribution, and intensity of the staining.

## 3. Results

The general health conditions of the animals remained stable throughout the experimental period, and all of them tolerated the experimental procedures well. There was no intragroup and intergroup difference in mean body weight throughout the experiment. On intraoral examination, no alterations were observed.

### 3.1. Histopathology

No bacterial contamination was observed except in two specimens in the MTA+pAsp group at 21 days, which were excluded from the analysis.

#### 3.1.1. Seven Days

All groups exhibited large areas of pulpal acute inflammation, which were intense in the control and MTA+pAsp specimens and moderate in the MTA and pAsp groups. Despite the altered pulpal condition, incipient formation of reparative dentin on the physiological dentin surfaces near the exposure site was detected in the treated groups. In contrast, the specimens showed no areas of reparative dentin below the exposure site (Figure 1A–D).

#### 3.1.2. Twenty-One Days

Pulpal inflammation was significantly reduced at this time point, except in the control group, where large necrotic areas surrounded by intense inflammatory infiltrate were evident. The specimens from the three treated groups showed extensive areas of normal dental pulp with numerous blood vessels, while the inflammatory infiltrate was observed only subjacent to the exposure site. The presence of a clear dentinal bridge in the three treated groups was a consistent finding (Figure 1E–H). Although the reparative dentin was thicker in the MTA and MTA+pAsp groups than in the 7-day groups, the one present in the pAsp specimens appeared in extensive areas, with fewer spaces and cell debris. The reparative dentin formed in specimens from the three treated groups showed no dentinal tubules, different from the adjacent physiological dentin (Figure 2A–C).

### 3.2. Immunohistochemistry

Due to the large areas of pulpal inflammation and the incipient and poorly delimited regions of reparative dentin observed at 7 days, immunohistochemical analysis was performed only in the 21-day specimens. As in the control group, the formation of reparative dentin did not occur; only the specimens of the treated groups were immunolabeled.

The reparative dentin observed in all treated groups showed intense immunostaining for OPN, which was evident both dispersed inside the matrix and at the bottom of the matrix (i.e., in the region of the secreting odontoblast-like cells). Physiological dentin showed no labeling for OPN (Figure 3A–C). On the other hand, the formed reparative dentin was immunoreactive for the DMP 1 antibody only in the pAsp specimens, in which it was dispersedly present throughout the matrix; in addition, labeling was also observed at the bottom of the matrix at the odontoblast-like cell layer. In specimens from the MTA and the MTA+pAsp groups, the reparative dentin matrix did not immunoreact for DMP 1. In addition, intense immunolabeling for DMP 1 appeared at the dentinal tubules from physiological dentin in all groups (Figure 3).

## 4. Discussion

The present immunohistochemical study showed the formation of a barrier of reparative dentin after experimental pulp exposure and capping in rat molars. Notably, the content of DMP 1 in addition to OPN in reparative dentin formed in the pAsp group suggested that the newly odontoblasts were fully differentiated, while the reparative dentin formed after capping with MTA and MTA+pAsp exhibited only immunopositivity for OPN.

The formation of a reparative dentin barrier involves the proliferation and differentiation of undifferentiated dental pulp cells to replace damaged odontoblasts. Calcium ions present in tissue fluid and released by pulp capping materials such as MTA participate in the cell-signaling process and, at a later stage, combine with phosphate ions present in tissue fluid to form hydroxyapatite crystals around the newly secreted collagen matrix [42,43]. The formation of liquid nanoprecursors sequestered by non-collagenous proteins is a fundamental step in biomineralization [29]. Several peptides, such as pAsp, are considered biomimetic analogs, as they are capable of forming Ca-P nanoclusters and initiating apatite formation into collagen matrices [34]. In fact, the formation of the mineralized barrier in the pAsp group, even in the absence of an additional source of calcium (MTA), confirms the high affinity of pAsp for Ca^2+^ ions from the physiological extracellular fluid. Obviously, from the clinical standpoint, the use of MTA presents advantages other than acting as a Ca^2+^ source, for instance, reducing bacterial contamination and sealing the exposure site by creating a thin necrotic layer [44]. These extra benefits, associated with the high cost of peptides such as pAsp, make the present results a “proof of concept” rather than an immediate alternative to conventional pulpal therapies.

At early stages, histopathological analysis of the treated specimens demonstrated larger areas of pulpal inflammation and degeneration, with incipient deposition of reparative dentin exhibiting a primary bone appearance characterized by the presence of trapped odontoblast-like cells in the mineralized dentin matrix [45]. The dental pulp in rats is highly reactive, and even in the absence of a pulp-capping agent, the formation of a mineralized barrier occurs due to a specific defense mechanism, which gives the pulp tissue greater resilience and healing capacity [46]. Similar to what was described in previous studies, the inflammatory reaction observed in the group treated with MTA was transient and occurred at levels that positively influenced pulp repair and the formation of a thick mineralized barrier [42,43]. Pulp exposures treated with pAsp showed a mild inflammatory response and necrotic areas in the pulp at 7 days. These changes were less severe than in the control group, and the pulp underlying the capped area was able to recover after 21 days, showing the formation of reparative dentin at similar levels to those achieved with the use of MTA. The stimulus to reparative dentin formation provided by the three treatment conditions was confirmed by the expression of OPN between the necrotic layer and the underlying pulp tissue and in the newly formed dentin, but not in physiological dentin, which indicates the specificity of immunolabeling [3,47,48]. OPN is a well-described high-phosphorylated sialoprotein resident of the mineralizing extracellular matrices of bone and cementum; it is also expressed in functional odontoblasts but secreted in small amounts into the dentin matrix [49]. This protein is characterized by the presence of a poly(aspartic) acid sequence and sites of Ser/Thr phosphorylation that mediate hydroxyapatite binding and a highly conserved RGD motif that mediates cell attachment/signaling. OPN expression is modulated by a large number of hormones, inflammatory cytokines, and growth and differentiation factors, such as epidermal growth factor (EGF), platelet-derived growth factor (PDGF), and transforming growth factor-β (TGF-β) [50]. In fact, the pAsp peptide tested in this study mimicked the poly(aspartic acid) domain present in the OPN sequence.

DMP 1 is secreted by fully differentiated odontoblasts and plays a fundamental role in dentinogenesis, mediating the mineralization of peritubular dentin and collagen fibrils from intertubular dentin [48,51,52,53]. Unlike previous investigations in which DMP 1 was expressed in response to calcium hydroxide and MTA [48,54], in the present study, DMP 1 was detected only in 21-day samples treated with pAsp, suggesting that only in this group were the odontoblast-like cells functionally mature. The faster cell differentiation can be attributed to the presence of pAsp in the exposure site, which would act on undifferentiated pulp cells, stimulating their migration, proliferation, and differentiation into odontoblast-like cells, upregulating the expression of proteins related to dentinogenesis, such as DMP 1 and promoting the deposition of endogenous calcium ions through chelation [22,23,25,55].

Differently from the direct pAsp application, the group treated with MTA modified with pAsp did not express DMP 1. It is possible that the substitution of 17 mass% (5:1 ratio) of MTA particles by pAsp negatively affected its setting reaction, as well as the characteristics of the set cement. In fact, increased setting time and solubility were observed in the modified MTA, which may have compromised the sealing of the exposure site, explaining the greater inflammatory infiltrate (caused by monomer leaching from the resin composite used to seal the cavity). Interestingly, the commercial form of pAsp contains significant amounts of its optical isomer (20% D-Asp) and its β structural isomer (71%), showing disordered conformation in mineralizing solutions containing high Ca^+^ concentration [56]. This elevated amount of β-bond structure may interfere with the poly-L-aspartic acid’s ability to form α-helix ordered secondary structures. Though analyses in the present study were conducted in demineralized samples, this conformational property could explain the distinct behavior between MTA+pAsp and pAsp groups on odontoblast differentiation. Finally, the alkalinization of the medium promoted by the MTA impairing the action of pAsp on the undifferentiated cells of the pulp is another hypothesis to the inferior results observed in the MTA+pAsp treatment group.

The results obtained in the present study are not without limitations. The formation of reparative dentin was followed for 21 days. Ideally, the analysis should have included longer periods to verify the presence of DMP 1 in the MTA-treated exposure sites. The lack of quantitative data as a limitation of the study is disputable. The main goal of immunohistochemistry is to identify the presence (or absence) of a given protein in a region of tissue and, most importantly, to identify its distribution. Thus, the absence of numerical data does not reduce the relevance of the main findings reported in this study.

## 5. Conclusions

Poly(aspartic) acid proved to be an alternative as effective as MTA in dentin regeneration in rats. Though all treatments led to the formation of reparative dentin after 21 days, only pulp exposures treated solely with poly(aspartic acid) showed positive staining for DMP 1, indicating its potential to direct stem cell differentiation towards the odontoblastic pathway. Therefore, the study hypothesis that the presence of pAsp at the pulp exposure site would affect the differentiation behavior of pulpal stem cells and the mineralization of the dentin matrix deposited by odontoblast-like cells was confirmed.

## Figures and Tables

**Figure 1 jfb-14-00537-f001:**
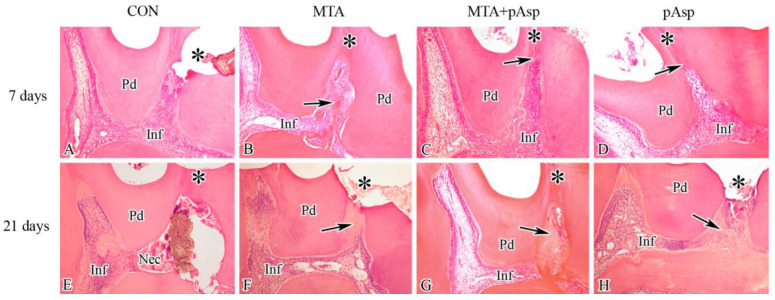
Light micrographs showing, at 7 days, the inflammatory infiltrate (Inf) in the control (**A**) and the treated (**B**–**D**) groups, as well as the incipient presence of reparative dentin (arrows) in the treated groups. At 21 days, areas of inflammatory infiltrate and pulpal necrosis (Nec) were present in the control specimens (**E**), while few inflammatory infiltrate areas were present in the dental pulp of treated groups (**F**–**H**), in which evident reparative dentin (arrows) was present. Thicknesses of the mineralized barriers are 118 μm (**F**), 452 μm (**G**), and 142 μm (**H**). Asterisks, regions of pulpal exposure/capping; Pd, physiological dentin. Hematoxylin-eosin staining. Original magnification: 100×. Scale bar = 100 µm.

**Figure 2 jfb-14-00537-f002:**
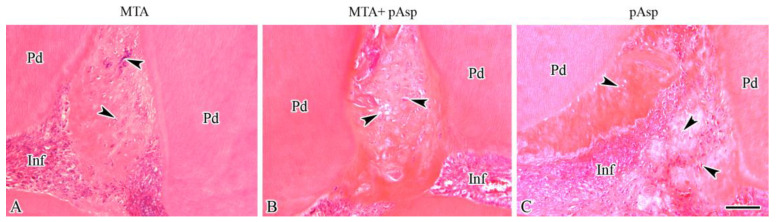
Light micrographs showing high magnification views of reparative dentin at 21 days. In all the treated groups, the irregular appearance of the reparative dentin matrix, which contains several cellular inclusions (arrowheads), was very different from the tubular aspect of the adjacent physiological dentin (Pd). Areas of inflammatory infiltrate (Inf) were observed subjacent to reparative dentin in specimen treated with MTA (**A**), as well as with MTA+pAsp (**B**) and pAsp alone (**C**). Hematoxylin-eosin staining. Original magnification: 200×. Scale bar = 50 µm.

**Figure 3 jfb-14-00537-f003:**
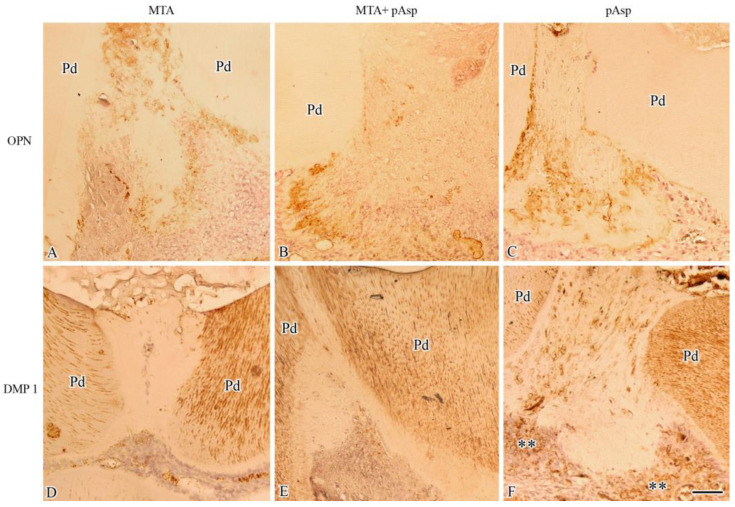
Light micrographs showing positive immunolabeling (in brown) for osteopontin (OPN) and dentin matrix protein 1 (DMP 1). OPN was identified in reparative dentin from the three treated groups (**A**–**C**), while for DMP 1, immunolabeling was negative for MTA (**D**) and MTA+pAsp (**E**) groups and positive only in the pAsp group (**F**). Immunoreactivity for OPN was absent in physiological dentin (Pd), while for DMP 1, it was positive at the dentinal tubules and in odontoblast-like cells (double asterisks). Hematoxylin counterstaining. Original magnification: 200×. Scale bar = 50 µm.

## Data Availability

Publicly available datasets are available for this study. This data can be found here: http://repositorio.uspdigital.usp.br/handle/item/553.

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
