# Peer review of "Poly(Aspartic Acid) Promotes Odontoblast-like Cell Differentiation in Rat Molars with Exposed Pulp"

_jfb, 2023, doi:10.3390/jfb14110537_

Round 1
Reviewer 1 Report
Comments and Suggestions for Authors
The topic of this study is interesting, but the all data is not objective but subjective. Thus, it is difficult to consider this manuscript to be a scientific article. The authors should digitize and analyze all data.
Comments on the Quality of English Language- Large problems are not recognized in English quality.
Author Response
Immunohistochemistry is a very specific approach by which a certain element from a cell or tissue is detected, in this case, the noncollagenous proteins osteopontin (OPN) and dentin matrix protein 1 (DMP 1) into the formed reparative dentin. The main goal of immunohistochemistry is to identify the presence (or absence) of a given protein into a region of tissue and, most important, to identify its distribution. Unfortunately, quantification methods are not advisable in this approach. For example, if one could attempt to quantify the intensity of labeling, it would be susceptible of changes, for example, with highly concentrating the antibody or extending the incubation time for the same slice. We have done immunohistochemistry (and even high-resolution immunocytochemistry, i.e., by electron microscopy, when the case is requiring!) without quantification for more than thirty years, with numerous publications.
In the present human study, the immunodetection of DMP 1 at the reparative dentin in the experimental group treated with pAsp, is a very specific finding. On the other hand, as expected, it was positive at the physiological dentin in all experimental groups. Thus, we consider that it is an objective rather than a subjective finding.
Reviewer 2 Report
Comments and Suggestions for Authors
TITLE: Poly(aspartic acid) promotes odontoblast-like cell differentiation in rat molars with exposed pulp
jfb-2582179-peer-review-v1
The aim of the present investigation was to to verify the effectiveness of poly(aspartic acid), pAsp, in dentin regeneration using an animal model.
GENERAL COMMENTS
The article is in-line with the journal topic, but flaws should be improved. The investigation is interesting, and the present paper is recommended for publication to the present journal after major revision.
Title: Poly(aspartic acid) promotes odontoblast-like cell differentiation in rat molars with exposed pulp.
Abstract
A structured form is strongly recommended. A brief sentence regarding the state-of-art background could be added.
Introduction
1. The author could describe the osteoinductive potential, bone-bonding ability (bioactivity) and bone biomineralization of current hydraulic calcium silicate cements used as graft materials and placed in contact with medullary bone (PMID: 28233601; DOI: 10.1016/j.dental.2017.01.017).
Materials and methods
1. The ethical approval statement and the conformity to the good clinical practice in animal research are missed.
2. The randomization methodology should be detailed.
3. In my opinion, the section could be empowered with histomorphometry evaluation.
4. The data analysis (and statistical methodology!) should be clearly detailed.
Results
The histopathology paragraph is too brief and should be detailed. No mentions about the vascularization of the specimens have been detected at 7 and 21 days. The indication of the histology magnification should be indicated in figures. A detailed description of the histological images should be added for both low and high magnification.
Discussion
In my opinion, the discussion section is well written, but the translational application of the study findings should be more accurate.
The clinical using of the studied cements should be detailed and discussed.
The limits of the present study and the rationale needs a precise disquisition.
Data availability
The link for the study data didn’t work. Please fix this issue providing the study meta-data.
The null-hypothesis should be discussed in this part of the manuscript.
Author Response
GENERAL COMMENTS
Abstract
A structured form is strongly recommended. A brief sentence regarding the state-of-art background could be added.
The abstract was modified as suggested.
Introduction
The author could describe the osteoinductive potential, bone-bonding ability (bioactivity) and bone biomineralization of current hydraulic calcium silicate cements used as graft materials and placed in contact with medullary bone (PMID: 28233601; DOI: 10.1016/j.dental.2017.01.017).
Thank you for the suggestion. A new paragraph was added to the introduction and the reccomended article was cited.
Materials and methods
The ethical approval statement and the conformity to the good clinical practice in animal research are missed.
This information can be found in the last paragraph of section 2.1. The Ethics Committee approval document was sent to JFB editorial office on August 14th 2023.
The randomization methodology should be detailed.
Thank you for the suggestion. The information was added on page 14.
In my opinion, the section could be empowered with histomorphometry evaluation. The data analysis (and statistical methodology!) should be clearly detailed.
We understand the reviewer’s concern regarding an additional histomorphometric analysis. However, the aim of our study was the immunodetection of the presence or absence of two noncollagenous matrix proteins, OPN and DMP 1, in the reparative dentin, as well as their localization. One strong reason for that was the current ethical restrictions on the use of animals in research. In the case of including some histomorphometric analysis, we would have had to add at least one more experimental period, maybe 45 days, in order to measure, for example, the thickness of reparative dentin and compare to that of 7 and 21 days. For that, the use of additional 28 rats would be necessary.
Results
The histopathology paragraph is too brief and should be detailed. No mentions about the vascularization of the specimens have been detected at 7 and 21 days.
More information was added to this section, as requested.
The indication of the histology magnification should be indicated in figures. A detailed description of the histological images should be added for both low and high magnification.
The original magnifications were added to the figure captions. Scale bars are shown in the leftmost image in each figure.
Discussion
In my opinion, the discussion section is well written, but the translational application of the study findings should be more accurate.
Two sentences were added in page 8, discussing the clinical viability of using pAsp in pulpal therapy.
The clinical using of the studied cements should be detailed and discussed.
The clinical uses of MTA are described in the new paragraph added to the introduction. At the authors’ discretion, the discussion focused on histological and immunohistochemical aspects.
The limits of the present study and the rationale need a precise disquisition.
A paragraph was added with the limitations of the study.
Data availability
The link for the study data didn’t work. Please fix this issue providing the study meta-data.
We double-checked the link. On the Word file, please click on the link while pressing the “Ctrl” key.
Conclusion
The null-hypothesis should be discussed in this part of the manuscript.
The confirmation of the null hypothesis was added to the conclusion.
Round 2
Reviewer 1 Report
Comments and Suggestions for Authors
The topic is interesting, but I'd like to recommend the data, the area of newly formed hard tissue for example, digitalized and semi-quantitated with image J.
Comments on the Quality of English LanguageNo problem.
Author Response
In order to obtain representative and relevant measurements, the histomorphometric analysis would have to include at least one more experimental period (for instance, 45 days). For that, the use of additional 28 rats would be necessary. Unfortunately, current ethical restrictions on the use of animals in research precluded that.
Notwithstanding, we decided to add the reparative dentin thickness measurements of the images in Figure 1 at 21 days to the caption.
Round 3
Reviewer 1 Report
Comments and Suggestions for Authors
The revised version of this manuscript can be accepted.